# Wave-based liquid-interface metamaterials

N. Francois[1], H. Xia[1], H. Punzmann[1], P.W. Fontana[2] & M. Shats[1]

The control of matter motion at liquid–gas interfaces opens an opportunity to create two-dimensional materials with remotely tunable properties. In analogy with optical lattices used in ultra-cold atom physics, such materials can be created by a wave field capable of dynamically guiding matter into periodic spatial structures. Here we show experimentally that such structures can be realized at the macroscopic scale on a liquid surface by using rotating waves. The wave angular momentum is transferred to floating micro-particles, guiding them along closed trajectories. These orbits form stable spatially periodic patterns, the unit cells of a two-dimensional wave-based material. Such dynamic patterns, a mirror image of the concept of metamaterials, are scalable and biocompatible. They can be used in assembly applications, conversion of wave energy into mean two-dimensional flows and for organising motion of active swimmers.

[1] Centre for Plasmas and Fluids, Research School of Physics and Engineering, The Australian National University, Canberra, Australian Capital Territory 2601, Australia. [2] Physics Department, Seattle University, 901 12th Avenue, PO Box 222000, Seattle, Washington 98122, USA. Correspondence and requests for materials should be addressed to M.S. (email: Michael.Shats@anu.edu.au).

Controlled manipulation of particles on a surface is important in a variety of material science and bio-engineering applications. Accordingly self-assembly, or the autonomous organization of components into patterns or structures, has become a fast growing interdisciplinary research area[1–3]. The controlled arrangement of particles on a liquid surface would offer a flexible and tunable method of engineering surface properties, such as, for example, electrical or thermal conductivity.

Recently new approaches to the manipulation of particles at a fluid surface have been proposed. They rely on the generation of surface waves to control the motion of particles at the liquid–gas interface. For instance, parametrically excited waves are generated when a liquid surface is vertically vibrated beyond a certain acceleration threshold. Such waves, often referred to as Faraday waves[4], are modulationally unstable[5] and are readily broken into ensembles of localised oscillating solitons, or oscillons[6,7]. In viscous liquids, oscillons can create spatially periodic patterns and it has been proposed that such patterns can be viewed as metamaterials[8]. The idea of employing Faraday wave patterns as templates for micro-scale assembly applications has recently been discussed[9]. Current understanding of what is achievable with those waves often relies on properties of particles such as their wettability and their density[10,11].

At lower fluid viscosity, steep oscillons form disordered wave fields which are coupled to a random motion of fluid particles at the surface[12,13]. This motion represents a macroscopic Brownian walk, where the diffusion of particles at the surface can be modified by changing the wave height and the wave length[14]. It has been shown that the properties of this interface are consistent with diffusion at thermal equilibrium, and as such, it can be viewed as a tunable thermal metafluid[15].

Another approach relies on propagating waves originating from a localized source to control the motion of fluid particles on the surface. In this case particles can be guided away from the wave source, as well as towards the source in the so-called 'tractor beam regime'[16]. In this case, as for Faraday waves, the waves are essentially nonlinear.

Here we show that there is another way of organising particles on a liquid–gas interface which is analogous to a method used in low-temperature physics to trap atoms in spatially periodic optical lattices[17]. We propose a method of remotely shaping the particle trajectories. It uses linear three-dimensional (3D) surface waves and relies solely on hydrodynamic forces. The main idea can be described as a combination of the concept of rotating waves, created by a superposition of two small-amplitude standing waves, and the Lagrangian drift of particles along closed paths in such waves. Though rotating waves (electromagnetic or acoustic) are usually generated in cylindrical resonators[18,19], we show below that periodic patterns of rotating waves can be created in a square geometry, similarly to optical lattices. On a fluid surface, such waves rotate within sub-wavelength cells and possess local angular momentum which is transferred to the matter. This mechanism produces particle trajectories in the form of a spatially periodic lattice of nested orbitals. This method offers a high degree of control over the particle motion and the ability to confine particles to spatially periodic cells. We present experimental results on the creation and control of such dynamical liquid metamaterials, develop a theoretical model of particle trajectories and characterize the transport properties of a multi-unit cell lattice.

## Results

**Experimental set-up**. Stationary surface waves are studied in a square container ($400 \times 400$ mm) filled with water to a depth of $d = 81$ mm. The waves are generated by two orthogonal horizontally oscillating paddles whose motion is driven by two electrodynamic shakers. The motion of the paddles (amplitude, acceleration) is accurately controlled via an accelerometer-based feedback loop (see 'Methods' section). In these experiments, we study linear waves of small amplitude ($H \approx 1$ mm $\ll \lambda$, where $H$ is the wave crest/trough amplitude and $\lambda$ is the wavelength). These waves obey the gravity-capillary wave dispersion relation: $\omega^2 = \tanh(Kd)(gK + \alpha K^3/\rho)$, where $\omega$ is the angular frequency of the waves, $K = 2\pi/\lambda$ is the wave number, $g$ is the gravitational acceleration, $\alpha$ is the surface tension and $\rho$ is the density of the liquid. Figure 1 shows schematically the experimental set-up (a,b) and a photo of the laboratory set-up (c,d).

In these experiments the frequencies of the paddle oscillations are chosen to fit an integer number of wavelengths into the square paddle-wall cavity ($312 \times 312$ mm$^2$). The relative temporal phase of the paddle oscillations can be tuned in the range of $\pm 180°$ with an accuracy of $\pm 0.1°$. This set-up allows the superposition of two planar standing surface waves to create a periodic wave field for which the relative temporal phase is controlled.

**Wave-driven fluid motion**. We study the motion of floating micro-particles on the water surface perturbed by surface waves. Two orthogonal plane standing waves create a 3D wave field as the one shown in Fig. 2a. First, we investigate trajectories of surface fluid particles tracked for one wave period $T = 2\pi/\omega$ at the nodal points. Nodal points are places on the surface where the local amplitude of the standing wave is zero at every instant in time. If the wave frequencies are equal, $\omega_1 = \omega_2$, the shape of the projection of the particle trajectory on the horizontal plane varies depending on the phase shift $\phi$ between the waves, as shown in Fig. 2b–d. A straight line corresponds to $\phi = 0$, a circle to $\phi = \pi/2$ and an ellipse to $\phi = \pi/4$. When $\omega_1 = 2\omega_2$, the trajectory is represented by a figure of eight (Fig. 2e). The projections of these trajectories on the horizontal plane are reminiscent of the Lissajous figures in a two-dimensional (2D) harmonic potential. In the case of an ideal irrotational fluid, the velocity field $\mathbf{u}^P = (u_x^P, u_y^P, u_z^P)$ associated with such trajectories can be represented by the gradient of a velocity potential $\Phi(x, y, z, t)$: $\mathbf{u}^P = \nabla\Phi$. The velocity potential in this wave field is given by:

$$\Phi(x,y,z,t) = A\cosh[K(z+d)][\cos(\omega_1 t)\cos(Kx) + \cos(\omega_2 t + \phi)\cos(Ky)], \quad (1)$$

where $K$ is the wave number, $d$ is the fluid depth and $A$ is the potential amplitude related to the wave amplitude $H$. The $z$-direction is upwards with $z = 0$ being the level of the undisturbed liquid surface.

The changes in trajectories observed in Fig. 2b–d highlight that both the wave dynamics and the trajectories of the fluid particles at the surface depend on the phase $\phi$. This can be seen in the temporal evolution of the normal $\mathbf{n}_f$ to the surface at a nodal point. Figure 2f,g shows the evolution of the horizontal projection of $\mathbf{n}_f$ for $\phi = 0$ and $\phi = \pi/2$ over one wave period. At zero phase shift, the temporal trace is a straight line, while it is a circle for the $\pi/2$ phase shift. This means that two orthogonal waves, phase shifted by $\phi = \pi/2$, possess local angular momentum. This angular momentum originates from the rotating character of standing waves, which is null at $\phi = 0$ and increases with $\phi$. This effect is somewhat surprising since equation (1) for standing waves produced in a square geometry shows decoupled temporal and spatial evolution. Indeed, this decoupling is manifest when $\phi = 0$. However, when $\phi = \pi/2$, a progressive rotating phase exists locally in the system. This can be described through a Taylor series expansion of $\Phi$ near a nodal point ($x_n, y_n, 0$), which reads:

$$\Phi(x,y,0,t) \approx -AK[\cos(\omega t)\sin(Kx_n)\delta x - \sin(\omega t)\sin(Ky_n)\delta y]$$
$$= \pm AK\,\delta r\cos(\theta \pm \omega t),$$

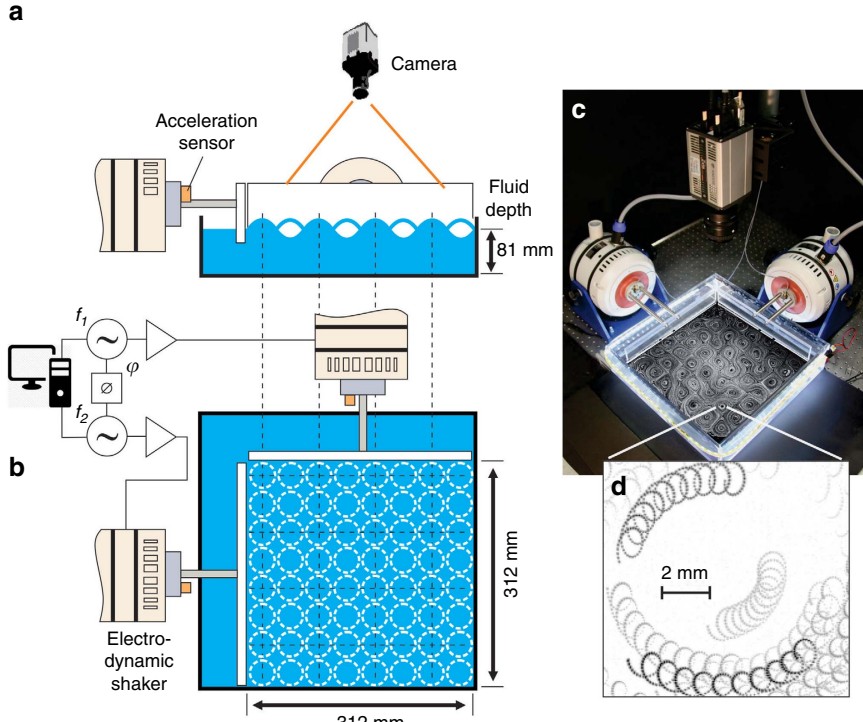

**Figure 1 | Experimental set-up.** (**a,b**) Schematics of the experimental set-up for the controlled superposition of two orthogonal standing waves in a fluid tank. Waves are created using two computer controlled electrodynamic shakers. The amplitudes, frequencies and relative phase of the two waves are adjusted with high accuracy. Both the wave field and the surface flow can be measured (see 'Methods' section for details). (**c**) A photo of the laboratory set-up showing the time-averaged streaks of drifting imaging particles. (**d**) Zoom into spatially resolved, small-scale particle drifting orbits.

where $\delta x = \delta r \cos \theta$, $\delta y = \delta r \sin \theta$ are expressed in polar coordinates $(r, \theta)$ centred at the nodal point and we note that $\sin(Kx_n) = \pm \sin(Ky_n) = \pm 1$. In this case a rotating phase $(\theta \pm \omega t)$ appears. This rotation is clearly seen in the experimental visualizations of the wave (Supplementary Fig. 1, Supplementary Movie 1).

Now we focus on the wave motion and particle trajectories when $\phi = \pi/2$. Figure 3a shows a snapshot of the wave topography measured at half the wavelength and marks the location of some remarkable points (peak, trough, nodal point, saddle points). First, the motion of the wave peak traces a square path, as illustrated by experimental data in Fig. 3b. The size of such a unit cell is $L_c = \lambda/2$. The motion of the wave extrema is discontinuous along the edges of this cell. In contrast, the $z = 0$ wave isoline rotates continuously at the frequency $\omega$ around the nodal point within the unit cell, as shown in Fig. 3b and in Supplementary Figs 1 and 3. The conservation of mass underpins the small-scale orbital motion of fluid particles at the time scale of the wave period $T$ (Fig. 2c). Here we show that a rotating wave can also transfer momentum to fluid particles. A slow drift of the orbits is observed in the direction of the wave rotation. This drift occurs along closed loops with a larger characteristic size ($\sim L_c$) and a large time scale (about $50T$) as seen in Fig. 3c. The direction of the orbital drift is opposite in adjacent unit cells and it follows the rotation of the wave (Fig. 3d).

**3D visualisation of travelling waves and the particle drift.** From a geometrical viewpoint, the existence of a small-scale gyroscopic motion coexisting with a slow drift motion is reminiscent of the Stokes drift observed in a planar progressive wave[20] (see the Theoretical Model section). This drift is revealed when a water wave is described from the Lagrangian perspective, or from the point of view of the fluid particle paths.

Experimentally, the Lagrangian nature of the particle drift can be illustrated by visualizing 3D trajectories of particles. Here we use a recently-developed method of 3D particle tracking on the surface perturbed by waves[16,21] with high spatial resolution ($\approx 10^{-3}\lambda$, where $\lambda$ is the wavelength) and high temporal resolution ($\approx 0.05T$). An example of a reconstructed trajectory is shown in Fig. 4a (red) along with its projection on the horizontal plane (green). Figure 4b and the Supplementary Movie 2 show instantaneous surface elevations at times $t_i$ superimposed on the trajectory with the position of the particle indicated by a small sphere. During half the wave period (high wave amplitude), the particle progresses in the direction of the wave rotation, while it moves backward during the second half. The particle's speed when the wave crest hits it, is higher than during the wave trough moment. This results in a small displacement of the particle in the direction of the wave propagation when a wave cycle is completed. Note that the physics of the Lagrangian circular drift revealed here is intrinsically different from a recent Eulerian theory of vorticity generation on a surface perturbed by waves, which considers bulk viscosity as the essential ingredient of the mechanism[22].

It should also be mentioned that we are able to generate such drift trajectories at the surface of various liquids with different viscosities (like glycerol–water solutions with viscosities in the range of $(1–10)$ mPa·s).

**Self-organised periodic lattice of rotating waves.** We now analyse the emergent properties of the fluid motion in a multi-unit cell lattice of such wave-driven metamaterial. Figure 5a–c shows particle streaks in flows produced at different phase shifts $\phi$ between two orthogonal standing waves. At $\phi = 0$ the flow is stationary but disordered. The flow becomes more ordered as $\phi$ increases.

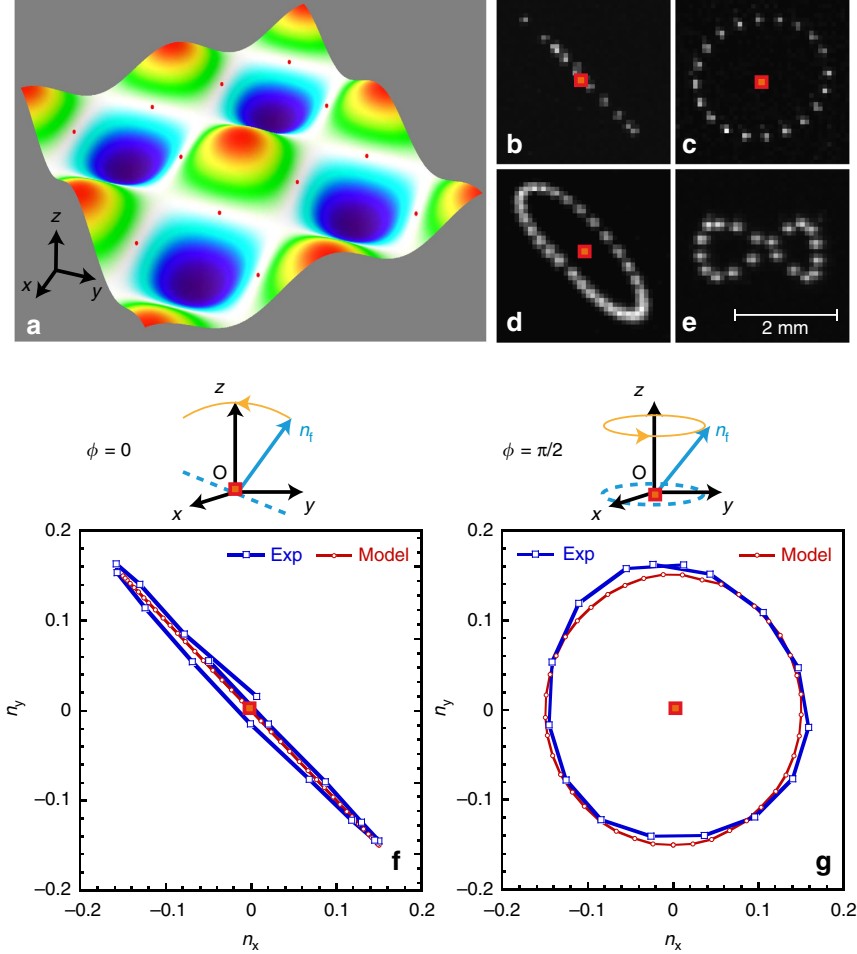

**Figure 2 | Surface elevation and surface particle orbits near the nodal points.** (**a**) Measured surface elevation produced by two orthogonal standing waves. Red dots indicate positions of nodal points. (**b**–**d**) Orbits of the surface particles near the nodal points (red squares) for different phase shifts $\phi$ between the standing waves: (**b**) $\phi = 0$, (**c**) $\phi = \pi/2$ and (**d**) $\phi = \pi/4$. The orbital motion is measured over one wave period $T = 2\pi/\omega \approx 0.26$ s. (**e**) A particle orbit near a nodal point for $\omega_1 = 2\omega_2$. (**f,g**) Temporal trace (over $T$) of the horizontal projections ($n_x$, $n_y$) of the water surface normals ($\mathbf{n}_f$ in the schematics) at a nodal point for **f** $\phi = 0$ and **g** $\phi = \pi/2$. Experimental measurements are compared with the theoretical model (see 'Methods' section) for $\omega_1 = \omega_2$, $f = \omega/2\pi = 3.9$ Hz ($\lambda = 104$ mm) and $H = 1$ mm.

First, we compute the compressibility of the flow in the horizontal ($x - y$) plane as:

$$C = \left\langle \frac{\left\langle (\partial u_x/\partial x + \partial u_y/\partial y)^2 \right\rangle_{x,y}}{\left\langle (\partial u_x/\partial y)^2 + (\partial u_y/\partial x)^2 + (\partial u_x/\partial x)^2 + (\partial u_y/\partial y)^2 \right\rangle_{x,y}} \right\rangle_{T_{av}} \quad (2)$$

Here $u_{x,y}$ denote fluid velocity components in the horizontal plane and $\langle \cdots \rangle_{T_{av}}$ denotes averaging over time $T_{av}$. Since the parameter $C$ is computed only on the horizontal components of the velocity, it characterizes the dimensionality of the flow. Quantitatively, $C$ can take on a value between 0 and 2. The lower the value, the more 2D is the flow. The value $C \approx 0.5$ marks the onset of 3D effects[23,24]. Figure 5d shows that $C$ is a decreasing function of $T_{av}$. For time scales comparable to the drift characteristic time ($\sim 50T$), the compressibility $C$ is actually much smaller than 0.5 independently of $\phi$ (Fig. 5d, inset). While the drift mechanism is a 3D phenomenon, the slow flow produced as a result of this drift is essentially 2D.

To characterize the fluid transport properties of the lattice we use the structure function $\rho(r) = \langle u(r_0) u(r_0 + r) \rangle_{r_0}/E$, where $E$ is the total horizontal kinetic energy of the surface flow. The angular brackets denote averaging over different positions $r_0$ in the flow of the products of horizontal speeds $u$ separated by a distance $r$. Figure 5e shows the Fourier transform of the structure function $\rho(k)$ (where $k = 2\pi/r$) for different relative phases $\phi$. At $\phi = 0$, the spectrum $\rho(k)$ spreads over low wave numbers ($k < 100\,\mathrm{m}^{-1}$) corresponding to the large-scale disordered streams seen in Fig. 5a. As the phase $\phi$ is increased, the broad distribution at low wave number is replaced with a strong peak at $k_w = 2\pi/L_C \sim 160\,\mathrm{m}^{-1}$. This peak characterizes the order emerging in the wave-guided transport at the spatial scale corresponding to the unit cell. While the phase $\phi$ is akin to a control parameter of the rotating wave momentum, the magnitude of the peak $\rho(k_w)$ can be viewed as a structure factor of the liquid-interface metamaterial. Figure 5f shows that $\rho(k_w)$ grows exponentially in the range $\phi = [0°, 40°]$ by a factor of about 20 and then saturates.

The emergence of order on the liquid surface with the increase in the phase $\phi$ is not the only strong effect. The total kinetic energy $E$ of the horizontal fluid motion increases with $\phi$, Fig. 5g, and for $\phi \geq 60°$ it is more than twice as large as that at $\phi = 0°$. Importantly, in the experimental data shown in Fig. 5, the wave amplitude produced by the oscillating paddles (that is, the energy injected in the system) is kept constant for all $\phi$. Thus the

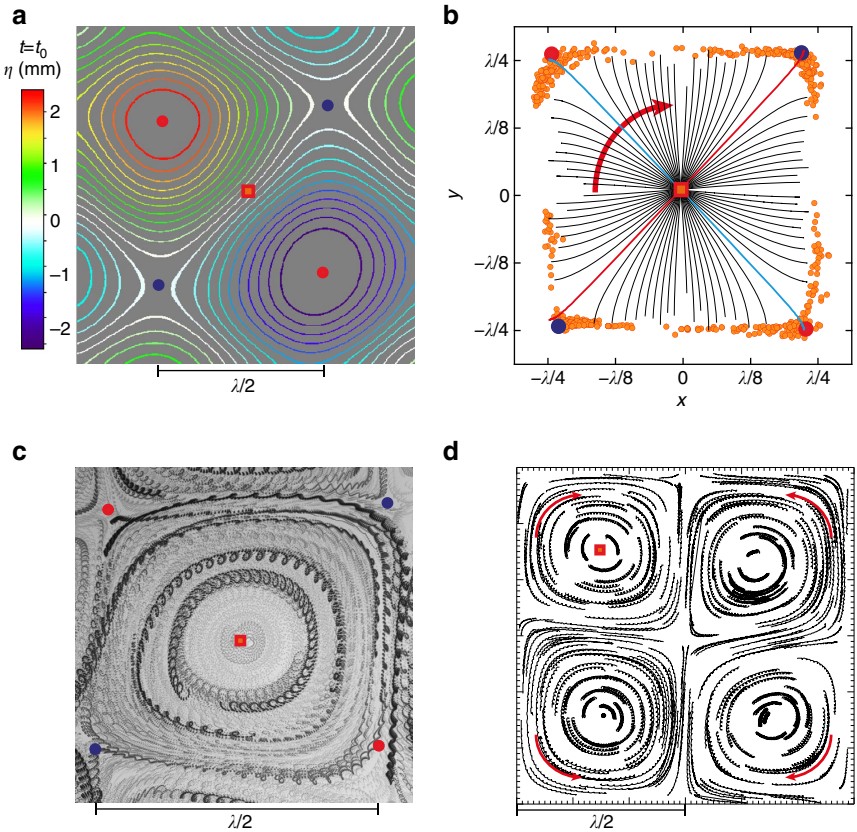

**Figure 3 | Surface topography and fluid particle trajectories at the surface for $\phi = \pi/2$.** (**a**) Contour plot of the surface elevation $\eta$ measured at the half-wavelength scale and $t = t_0$. The positions of a nodal point (red square), peaks/troughs (red circles) and saddle points (blue circles) are highlighted. (**b**) Dynamics of the rotating wave about a nodal point within a unit cell of size $L_c = \lambda/2$. Orange circles: motion of wave peaks experimentally tracked for 50T. Black lines: the rotation of the $z = 0$ isoline of the surface elevation followed for $T/2$. (The red line indicates $t = t_0$, the blue one $t = t_0 + T/4$.) (**c,d**) Surface particle drifts tracked for $\approx 50T$: (**c**) within a single unit cell, particle orbits drift forming closed nested guiding centre trajectories (experiments, $f = \omega/2\pi = 3.9$ Hz ($\lambda = 104$ mm), $H = 2.5$ mm). (**d**) The direction of the drift alternates in adjacent unit cells (experiments, PTV measurements, $f = 3.9$ Hz, $H = 1$ mm).

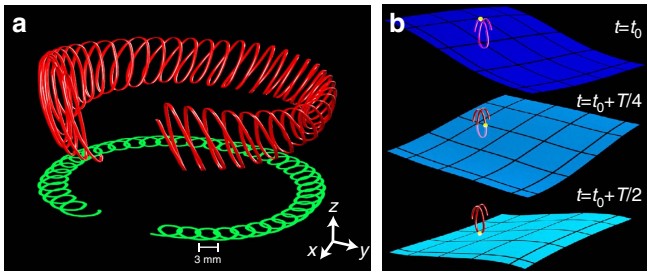

**Figure 4 | Rotating drift mechanism.** (**a**) Experimentally measured 3D trajectory (red) of a surface particle drifting within a unit cell and its projection on the horizontal plane (green) (experiments, $f = \omega/2\pi = 3.9$ Hz, $\lambda = 104$ mm, $H = 2.5$ mm). (**b**) Positions of an imaging particle (yellow dot) on the 3D trajectory at three consecutive moments in time within half a wave period. See Supplementary Movie 2 for details. Blue surfaces show the rotation of the liquid surface measured simultaneously with the particle position.

emergence of rotating waves at the unit cell scale increases the conversion rate of the wave vertical oscillatory energy into spatially ordered horizontal kinetic energy.

**Theoretical model of the 3D rotating drift.** As mentioned above, it is tempting to draw an analogy between the 3D rotating drift and the 2D Stokes drift for planar progressive waves[20]. In the classical picture uncovered by Stokes, the Eulerian velocity at a point averages to zero over one wave period, Fig. 6a, but the Lagrangian velocity of a fluid particle gyrating in the

neighbourhood of that point does not, such that the fluid particle drifts in the direction of the wave propagation (Fig. 6b).

To test the relevance of such an analogy, we use the incompressible Euler equations to model the flow $\mathbf{u} = (u_x, u_y, u_z)$ and the liquid surface $\eta = \eta(x, y, t)$ of an ideal fluid perturbed by orthogonal standing waves. Such equations read:

$$\frac{\partial \mathbf{u}}{\partial t} + \mathbf{u} \cdot \nabla \mathbf{u} = -\frac{1}{\rho} \nabla p \qquad (3)$$

$$\nabla \cdot \mathbf{u} = 0 \qquad (4)$$

with the following boundary conditions:

$$u_z|_{z=-d} = 0 \qquad (5)$$

$$\left[\frac{\partial \mathbf{u}}{\partial t} + \mathbf{u} \cdot \nabla \mathbf{u}\right]_{z=\eta} = -g\nabla\eta \qquad (6)$$

$$\frac{\partial \eta}{\partial t} + \mathbf{u} \cdot \nabla\eta = u_z|_{z=\eta} \qquad (7)$$

where $z$ is measured upwards from the unperturbed liquid surface and $\rho$ is the fluid density. The modified pressure $p$ includes the gravitational pressure $\rho g z$ (constant atmospheric pressure is assumed at $z = \eta$). Equation (5) imposes zero flow through the rigid bottom of the container. Equation (6) expresses horizontal momentum balance at the surface ('dynamical free surface boundary condition'). Equation (7) relates changes in the surface displacement to the vertical component of the flow ('kinematic free surface boundary condition').

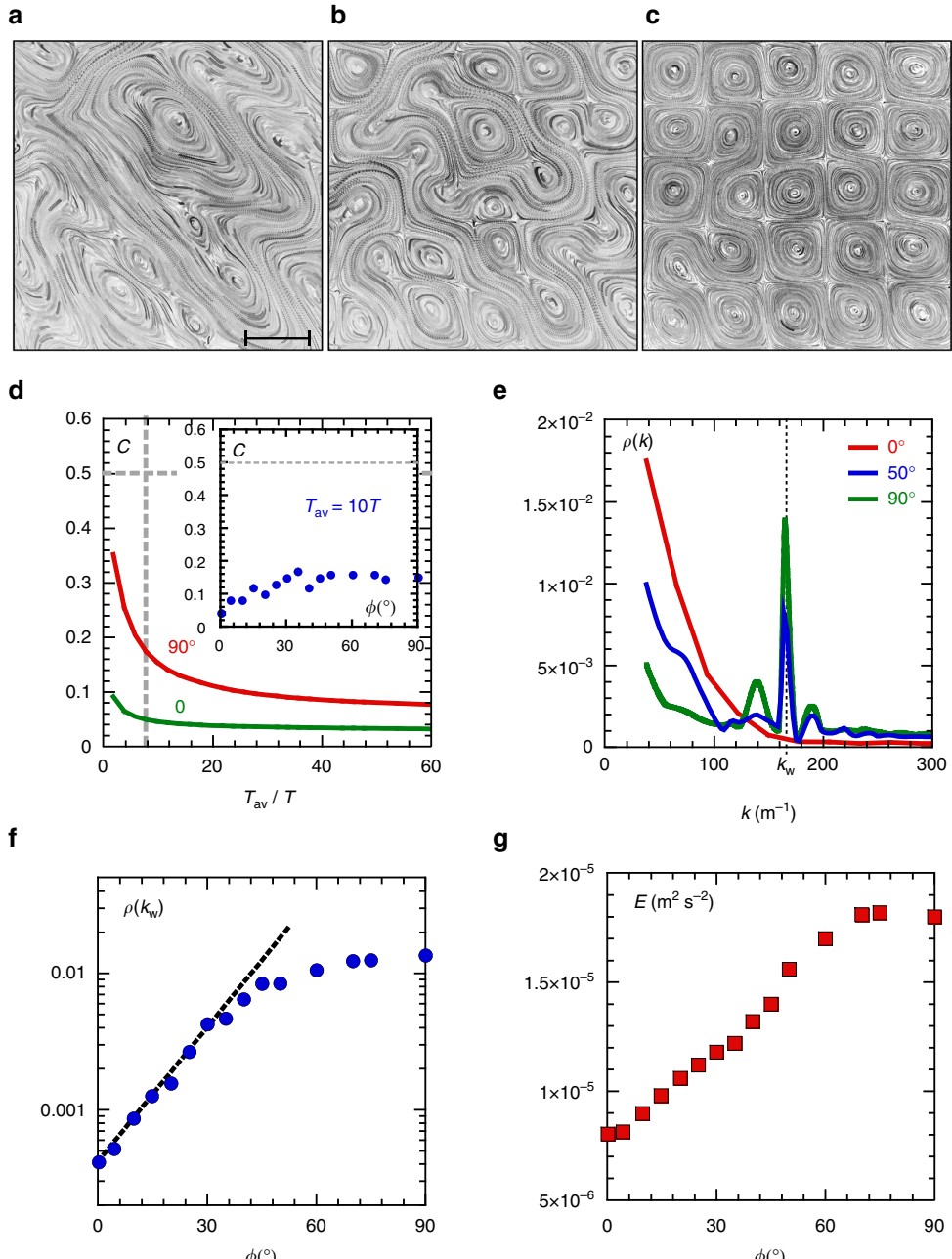

**Figure 5 | Liquid-interface metamaterial.** (a–c) Surface particle streaks measured at different phases $\phi$: (a) $\phi = 0$, (b) $\phi = 50°$, (c) $\phi = 90°$ (experiments, $f = \omega/2\pi = 4.58$ Hz, $\lambda = 78$ mm, $H = 2$ mm, $5 \times 5$ unit cells shown out of the $8 \times 8$ lattice formed in the cavity; Scalebar, 39 mm.). (d) Compressibility $C$ of the horizontal flow, equation (2), measured at $\phi = 0$ (green) and at $\phi = 90°$ (red) versus averaging time $T_{av}$ normalized by the wave period $T$. Inset: $C$ averaged over 10 wave periods is small, $C < 0.2$ in the range of $\phi = (0 - 90)°$. (e) Wave number spectrum of the structure function $\rho(k)$ for different relative phases $\phi$. As $\phi$ approaches $\pi/2$, the flow develops spatial order, indicated by a peak at $k_w$ corresponding to $\lambda/2$. (f) The onset of the spatially ordered flow is seen as an exponential growth of $\rho(k_w)$ with $\phi$ in the range of $\phi = (0 - 40)°$. (g) The mean horizontal kinetic energy $E$ of the surface flow increases with $\phi$ by a factor of $> 2$.

Under these conditions, the velocity field $\mathbf{u}^P = \nabla\Phi$ (equation (1)) is a solution. Assuming the phase difference $\phi = \pi/2$, we obtain:

$$\mathbf{u}^P = \frac{LKg}{\omega} \frac{\cosh[K(z+d)]}{\cosh Kd} \{ -\cos\omega t \sin Kx\,\hat{\mathbf{x}} + \sin\omega t \sin Ky\,\hat{\mathbf{y}} + \tanh[K(z+d)](\cos\omega t \cos Kx - \sin\omega t \cos Ky)\hat{\mathbf{z}} \} \quad (8)$$

where $L \equiv A\omega\cosh(Kd)/g$ has the dimension of a length. Lagrangian particle trajectories can be computed by numerical integration of equation (8) with $z = \eta$. The resulting trochoid-like

trajectories circulate in nested orbits about the unit cell in the direction of the rotating wave (Fig. 6c,d) in agreement with the experimental observations. A given trajectory exhibits small-radius gyrations at frequency $\omega$ superimposed with a circular drift about the unit cell at a much lower frequency. Analytically, the drift velocity $\mathbf{U}_d$ can be computed as:

$$\mathbf{U}_d = \overline{\left( \int_0^t \mathbf{u}^P dt' \right) \cdot \nabla\mathbf{u}^P}, \quad (9)$$

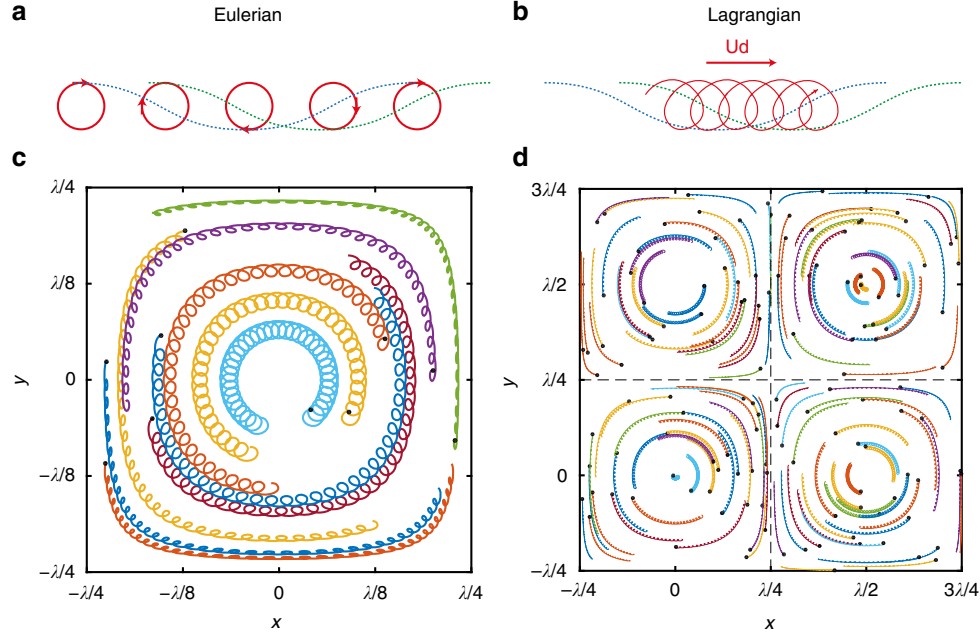

**Figure 6 | Modelled fluid particle trajectories at the fluid surface perturbed by waves.** (**a,b**) Fluid particle motions (red curves) in a 2D linear wave propagating from left to right (blue and green dashed lines). (**a**) At order $L$ in the particle displacement (see equations (8 and 10)), the instantaneous Lagrangian and Eulerian velocities are identical, the particle paths are closed loops. The period average velocity of the five trajectories shown is null. (**b**) Same linear wave, the fluid motion is now described (at order $L^2$) in the Lagrangian frame: a particle follows a trochoidal curve, which illustrates the period-averaged Stokes drift velocity $\mathbf{U}_d$. (**c,d**) Modelled surface particle trajectories in 3D linear standing waves for $\phi = \pi/2$. This modelling corresponds to the experimental data shown in Fig. 3c,d. Surface particle drifts are tracked for $\approx 50T$, the parameters are $f = \omega/2\pi = 3.9\,\text{Hz}$ ($\lambda = 104\,\text{mm}$), $H = 2.5\,\text{mm}$ in (**c**) and $f = 3.9\,\text{Hz}$, $H = 1\,\text{mm}$ in (**d**). These Lagrangian particle trajectories computed by numerical integration of the Eulerian equation (8) exhibit small-radius gyrations at frequency $\omega$ superimposed with a circulatory drift about the unit cell at a much lower frequency. The black dots signal the initial positions of the particles. The direction of the simulated drift alternates in adjacent unit cells.

where $\overline{(\cdots)} = \frac{1}{T}\int_0^T (\cdots)\mathrm{d}t$ is a gyro-average over one gyration period $T = 2\pi/\omega$ (note that $\mathbf{U}_d$ is a time-averaged Lagrangian property and $\mathbf{u}^P$ is an Eulerian velocity). The drift speed $U_d$ approximately matches the experimentally measured one. Counter-intuitively, a quasi-linear irrotational model can produce vortex-like structure at a fluid interface[25,26]. For more quantitative agreement, we include an additional steady rotary motion at the surface to take into account the finite steepness of the waves (see 'Methods' section). The existence of such steady rotation has also been demonstrated in numerical simulations of non-linear waves at the surface of an ideal fluid[27,28].

## Discussion

Our results show that a horizontal flow on a liquid surface can be ordered into a perfect lattice of counter-rotating vortex-like structures by adjusting the temporal phase between two orthogonal surface waves to $\pi/2$. Such vortices confine floating particles within the unit cells. Manipulation of matter using waves is well known in a range of physical contexts[29–33]. It is particularly interesting to compare trapping of fluid particles within unit cells of the surface-wave metamaterial with the case of optical lattices in which atoms are trapped in the potential landscape created by two standing optical waves[29,30]. In optical lattices, the formation of radiation pressure vortices has been reported[17,31]. Radiation pressure is related to the momentum of electromagnetic waves, and the vortices are generated when two orthogonal standing waves have their temporal phases shifted by $\pi/2$, similar to the case of the surface waves. The wave energy flows along closed paths that resemble fluid vortices and the radiation force acting on the atoms is non-conservative[31,32].

Although such an analogy between optical lattices and the surface-wave-based metamaterial is remarkable, the two systems are quite different.

Surface waves are strongly dispersive and they obey mass conservation in the fluid. In addition, waves in vacuum and waves in a medium have other fundamental differences. There is usually no clear connection between a mechanical wave and the momentum it may generate in a fluid[34–36]. Examples include an acoustic linear wave which can propagate in a fluid without any momentum present, or a small gravity wave packet which can generate pressure disturbances (that is, momentum) in a fluid far away from the packet spatial location[34].

In this context, the Stokes drift has a special role: it can be viewed as a wave momentum[34]. Indeed, the phenomenon intimately links the transport of surface fluid particles to a propagating planar surface wave. Here we uncover a wave configuration where the Stokes drift exists along a closed path: a rotating Stokes drift. This path is created by a locally rotating wave. In this sense, the results presented here point to the existence of a radiation–pressure-like force guiding particles at the liquid surface.

It would also be interesting to find an analogue of a particle polarizability by light waves in the context of water waves. In optics, the particle polarizability plays a major role in both the generation of optical vortices and in trapping of atoms. Similarly to the case of optical traps, it was shown that in the standing surface waves, particles' inertia and wettability conspire to trap particles either at the wave extrema or at the nodes[10]. This suggests that the analogue of the polarizability in the surface waves could be related to the wettability and inertia of particles. In our experiments, the rotating Stokes drift clearly dominates the trapping effects.

Beyond these analogies, the scope of the recently introduced concept of a wave-driven metafluid[15] can be broadened to that of a wave-based liquid-interface metamaterial. As we show, by dynamically shaping a fluid interface using rotating waves, one can produce an effective 2D material endowed with prescribed transport properties. An important feature of such a system is the establishment of unit cells confining nested spatio-temporal structures. These structures would allow remote control of particles on multiple length scales.

## Methods

**Experimental set-up.** Figure 1 shows schematically the experimental set-up (a,b) and a photo of the laboratory set-up (c). Computer controlled electrodynamic shakers (TIRA TV51140) are used to drive synchronized motion of the two wave paddles. The paddle accelerations are measured using two accelerometers (B&K 4507, 1,000 mV g$^{-1}$) that provide feedback to the system's motion controller (Vibration Research, VR9500). The phase delay $\phi$ between the paddles is adjustable in the range of $\pm 180°$ via an arbitrary waveform generator (HP 33120 A).

Particular attention was paid to boundary conditions at both the oscillating paddles and the fixed wall facing them. It was noticed that the presence of a meniscus strongly influences our ability to produce a well-controlled standing wave field. Indeed we observe that a meniscus affects the spatial control on the phase $\phi$ in the cavity, and it was recently shown to affect the reflection coefficient of gravity-capillary waves on the wall[37]. Therefore we machine specific grooves in the container walls and paddles such that the contact line is pinned to the wall edge with no meniscus. The wave fields produced in this cavity match well the numerically modelled waves (see Supplementary Figs 1 and 3).

**Flow measurements.** The horizontal fluid flow is visualised using buoyant tracer particles (Polyamid, 50 μm) illuminated by light-emitting diode strip lights surrounding the transparent acrylic fluid tank. A high-resolution video camera (Andor Zyla X5.5; 2,560 × 2,160 pixel; 100 fps) is used to film the motion of imaging particles. With 16-bit resolution, the camera provides sufficient pixel intensity and spatial resolution (∼50–200 μm per pixel; Nikon f1.4/50 mm lens) for quantitative data analysis using well-developed particle imaging velocimetry and particle tracking velocimetry (PTV) algorithms.

The particle imaging velocimetry technique is used to obtain velocity fields of the horizontal fluid motion. During an experimental run, the particle motion (corresponding to different $\phi$) is recorded twice: once at a frame rate equal to the wave frequency $f = \omega/2\pi$ and then at $(10 \times f)$ fps. This allows us to characterize accurately both the fast gyration motion and the slow rotating drift. For the data presented in Fig. 5, the field of view used for the analysis is 234 × 234 mm$^2$ corresponding to a 6 × 6 unit cell lattice (for waves generated at $f = \omega/2\pi = 4.58$ Hz, $\lambda/2 = \pi/K = 39$ mm). The spatial resolution is 125 μm per pixel. The velocity fields are computed on a 40 × 40 spatial grid (grid mesh size is $\approx 5.35$ mm), with a 10.7 × 10.7 mm$^2$ interrogation window size (the interrogation windows are overlapping). The measurement resolution of the instantaneous displacement is subpixel. The wave number spectra in Fig. 5e are averaged over the field of view and 200 snapshots of the velocity field.

We use diffusive light imaging to measure the wave motion. The fluid surface is illuminated using a light-emitting diode panel placed underneath the transparent bottom of the container. A few per cent of milk added to water provides sufficient contrast to obtain a high-resolution reconstruction of the wave field. The absorption coefficient is measured before each experiment, which allows calibrating the surface elevation with a vertical resolution of 80 μm. For 3D PTV measurements, floating black carbon glass particles are used to visualise simultaneously the fluid and the wave motions. A few drops of non-ionic surfactant are added to ensure that these particles are homogeneously distributed at the fluid surface before starting an experimental run. For a given set of parameters (paddles acceleration, frequency), no difference could be measured between particle horizontal motion in these experiments and the trajectories of Polyamid particles at the surface of water with no milk and no surfactant.

3D Lagrangian trajectories (see Fig. 4, Supplementary Movie 2 and Supplementary Fig. 2) are retraced using a combination of a 2D PTV technique and a subsequent estimation of the local elevation along the trajectory[21]. First, the horizontal motion ($x - y$ coordinates) of a particle is tracked using threshold filters and a nearest-neighbor algorithm[38]. The wave field evolution is then obtained by removing particles from the movies with local filter techniques. Then the particle elevation ($z$ coordinate) is measured as the wave elevation over a local window (300 μm radius), which is centred on the $x - y$ particle coordinates at a given time. Finally the 3D trajectories of the particle and the wave field are visualized using the Houdini 3D animation tools (Side Effects Software).

**Theoretical effect of the finite wave steepness.** Analytically, the drift $\mathbf{U}_d$ (equation 9) associated with the Eulerian velocity (equation 8) is

$$
\begin{aligned}
\mathbf{U}_d &= \overline{\left(\int_0^t \mathbf{u}^P \mathrm{d}t'\right) \cdot \nabla \mathbf{u}^P} \\
&= \frac{L^2 g^2 K^3}{2\omega^3}\frac{\sinh^2[K(z+d)]}{\cosh^2(Kd)}\left(-\sin Kx \cos Ky\,\hat{\mathbf{x}} + \cos Kx \sin Ky\,\hat{\mathbf{y}}\right),
\end{aligned}
\tag{10}
$$

where $\overline{(\cdots)} = \frac{1}{T}\int_0^T (\cdots)\mathrm{d}t$ is a gyro-average over one gyration period $T = 2\pi/\omega$. This expression agrees quantitatively with the gyro-averaged velocity observed by numerical integration of the Eulerian equation (8). The parameters $L$, $K$, $\omega$ and $d$ are taken from the experiment.

The small parameter $K\eta$ is a measure of the steepness of the wave and is much less than 1 for surface vertical displacements of several millimetres across a 4 cm-wide unit cell as considered here. In our modelling, we have also considered the wave/flow coupling to first order in $\mathcal{O}(K\eta)$. The modelled surface displacement $\eta$ comes from equation (6) with $\mathbf{u} = \mathbf{u}^P$. To find $\eta$, we substitute equation (8) into equation (6), invert the gradient and retain terms of order $\mathcal{O}(K\eta)$. The result is:

$$
\eta = \frac{\eta_1 + \eta_2 + \eta_3 + \eta_4 + \tanh^2(Kd)\eta_5}{1 - \tanh(Kd)K(\eta_1 + \eta_2 + 2\eta_5)},
\tag{11}
$$

where

$$
\eta_1 = L\sin\omega t\cos Kx, \quad \eta_2 = -L\cos\omega t\cos Ky
$$

$$
\eta_3 = \frac{L^2 gK^2}{4\omega^2\cosh^2(Kd)}\cos^2\omega t\cos 2Kx, \quad \eta_4 = \frac{L^2 gK^2}{4\omega^2\cosh^2(Kd)}\sin^2\omega t\cos 2Ky
$$

$$
\eta_5 = -\frac{L^2 gK^2}{2\omega^2}\sin 2\omega t\cos Kx\cos Ky.
$$

Defining $H$ as the maximum surface displacement, we find that $H \approx \sqrt{2}L$ when $KL \ll 1$. For a time sequence of $\eta$ for parameters relevant to the experiment ($L = 1.3$ mm, $K = 58.1$ rad m$^{-1}$, $\omega = 23.9$ rad s$^{-1}$, $d = 8$ cm) and comparison with experimental data, see the Supplementary Figs 1 and 3 and Supplementary Movies 1 and 3. There is a strong match between the experimentally measured wave field dynamics and the predictions of the model.

To take into account higher-order effects of the finite steepness of the waves on the flow, we add to the model a rotational flow $\mathbf{u}^R$ in addition to the intrinsic Lagrangian drift. Specifically:

$$
\begin{aligned}
\mathbf{u}^R &= b\big[-\sin Kx \cos Ky\,\hat{\mathbf{x}} + \cos Kx \sin Ky\,\hat{\mathbf{y}}. \\
&\quad + \frac{b^2 K}{g}\sin 2Kx \sin 2Ky(\cos 2Kx - \cos 2Ky)\hat{\mathbf{z}}\big]
\end{aligned}
\tag{12}
$$

where $b$ is a rotational velocity amplitude. The horizontal components of $\mathbf{u}^R$ are chosen to match the form of the drift velocity $\mathbf{U}_d$. The introduction of such a term is supported on physical grounds. Indeed we have noted that the model (11) at order $\mathcal{O}(K\eta)$ predicts a time-averaged surface gradient outward from the centre of each unit cell, that is, a stationary well, providing a centripetal acceleration within the unit cell. A time-averaged remnant level profile has been discussed in the context of the effect of radiation pressure on 2D standing waves[35]. On a related vein, we note that a horizontal drift of a similar form was mentioned in a nonlinear analysis of non-viscous standing surface waves[27]. When the rotation amplitude b is chosen to match the experimental data, $\mathbf{u}^R$ is of the order of magnitude of the drift $\mathbf{U}_d$ (see Supplementary Fig. 3 and Supplementary Movie 4).

The particle trajectories shown in Fig. 6c,d were produced by integrating the velocity $\mathbf{u} = \mathbf{u}^P + \mathbf{u}^R$ using a 3D, fourth-order Runge–Kutta algorithm. The model parameters used were $L = 1.3$ mm and $b = 0.8456$ cm s$^{-1}$ for **c** and $L = 0.433$ mm and $b = 0.4882$ cm s$^{-1}$ for **d**. The trajectories were integrated for 2,500 points in time over 37 gyration cycles (c) and 25 gyration cycles (d).

**Data availability.** The data that support the findings of this study are available from the corresponding author upon reasonable request.

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

## Acknowledgements

This work was supported by the Australian Research Council's Discovery Projects funding scheme (DP150103468 and DP160100863). H.X. acknowledges support from the Australian Research Council's Future Fellowship (FT140100067). N.F. acknowledges support by the Australian Research Council's DECRA award (DE160100742).

## Author contributions

N.F., M.S., H.X. and H.P. designed and performed experiments and analysed the data. P.W.F. developed the theoretical model. H.X. and P.W.F. performed numerical simulations. N.F., M.S and P.W.F. wrote the paper. All authors discussed and edited the manuscript.

## Additional information

**Competing financial interests:** The authors declare no competing financial interests.

**Publisher's note**: 

