## [Peer Review File · Nature Communications]

Reviewers' comments:

Reviewer #1 (Remarks to the Author):

The authors present a very interesting approach to direct the motion of small particles on a liquid surface by sculpting wave-controlled patterns of surface flows. In particular, they have been able to create periodic patterns of rotating waves that can transfer local angular momentum to micro-particles leading to micro-particle trajectories in the form of spatially periodic nested orbitals. I found the experimental (and theoretical) results very exciting and the work certainly merits for publication in NCOMMS after some mandatory revisions that are summarized below:

1) The authors state in the introductory paragraphs: "Here we show that a liquid surface metamaterial can be created by wave-controlled patterns of surface flows. The attraction of this wave approach is that it relies solely on hydrodynamic forces to guide particles". The authors already discussed the same idea in Ref. 21 where they introduced "a new conceptual framework for understanding wave-driven flows". The authors should clarify and emphasize the actual novelty and relevance of their results compared with their previous work.

2) In the abovementioned reference [21] the authors used wave-controlled patterns of surface flows to develop a surface-wave analogue of an Optical Tractor Beam [It is interesting to note that while in the ArXiv version of this paper, the authors cited and explicitly acknowledged that the work was motivated by previous work on Optical Tractor Beams, in the published version in Nature Physics (Ref. 21), references to the previous work on optical forces were removed...]. In the present work, the authors discussed the flow patterns associated to superposition of orthogonal standing waves. The optical analogue of the author's approach can be found in an early work on optical lattices by Hemmerich and Hänsch [A. Hemmerich and T.W. Hänsch, Phys. Rev. Lett. 68, 1492 (1992)] and the flow patterns of nano-particles illuminated by two crossed optical standing waves have been discussed by Saenz and coworkers [Albaladejo et al., Phys. Rev. Lett. 102, 113602 (2009); Nano Lett. 9, 3527 (2009); Zapata et al., Phys. Rev. Lett. 103, 130601 (2009).] These references should be cited in the manuscript.

3) It is remarkable that, in full analogy with the Optical case, when there is a $\pi/2$ phase shift between the orthogonal standing waves, the flow patterns resemble those of a periodic vortex lattice. While the main driving forces in the optical case are proportional to the energy flow (the Poynting vector), the dynamics of the particles on the surface of the fluid, driven by the fluid flow, is more complex.

Could the authors include a brief discussion of the similarities and differences of the driving by light radiation pressure and liquid flow?

This will help the interdisciplinary readers of NCOMMS to understand better the beautiful results presented by the authors.

Reviewer #2 (Remarks to the Author):

Combining experiments and theory Francois et al demonstrate and explain the orbital and drifting motions of solid particles surfing on Faraday-wave lattices.

The experiments are very elegant and yield extremely accurate results both for the interface and the (3D) particle dynamics. They rely on a host of advanced techniques mastered by the authors for the interface and flow measurements. The model is simple and quantitatively accounts for most of the experimental findings. However I cannot recommend this manuscript for publication in the present form: (i) The presentation of the results is often unclear, at least to nonspecialists in the field. I would strongly recommend adding to the main text some of the explanations provided in the method section. This comments especially applies to the theoretical model and to the description of the experimental setup. (ii) The main text is marred with unsubstantial statements

about potential applications of this experiment. I feel that this should be rectified. (iii) The following more specific points should also be addressed:

-1- Unlike what is suggested in the abstract, the dynamic control of particles by wave fields is very common. The most obvious example is provided by the optical lattices routinely used in atomic physics and colloidal science.

-2- A lot of emphasis is put on the manipulation of colloidal particles, yet the scale of the lattice used in this experiment is of the order of a centimeter. Positioning colloids with a millimeter precision on a square lattice with a spacing of 1 cm would be definitely easier using pipets. Moreover, capillary forces would definitely dominate the Stokes drag driving the particles. The authors should either quantitatively address these points or significantly adjust their enthusiasm about the relevance to colloid and cell manipulation.

-3- What is specific to the nonlinearity of the Farady waves? Again, I am not an expert in interfacial waves and Lagrangian transport, reading this manuscript I could not understand what could not be achieved with standing capillary or gravity waves. Would the physics and the phenomenology be qualitatively different?

-4- The description of Eq. (2) is not correct. ξ is the particle displacement, not its instantaneous position. Galilean invariance would be broken otherwise... Eq. 2 also includes a mean flow term (which vanishes in the present case)

-5- Some of the panels in figures 1 and 3 are not discussed in the main text.

-6- Given their importance a clear definition of the nodal points is required.

-7- When does particle inertia becomes relevant (particle size and oscillation frequency)?

-8- The theoretical explanation for the rotation drift term should be explain in more details. This is one of the main result of the experiment, as far as I understand it, Eq. 12 seems to be established on a sole phenomenological basis. Is it correct?

-9- I found the discussion of the "collective" properties rather confusing. What is collective here? As far as I understand this last section, only the flow field is characterized, not the particle positions. The particles are used as passive tracers, and do not interact. I do appreciate the characterization of the non linear response of the flow, but do not see any sign of a collective phenomena.

-10- The authors might want to refer to this recent paper; Fluids by design using chaotic surface waves to create a metafluid that is Newtonian, thermal, and entirely tunable, by Welch et al, P.N.A.S 2016.

Reviewers' comments:

Reviewer #1 (Remarks to the Author):

The authors present a very interesting approach to direct the motion of small particles on a liquid surface by sculpting wave-controlled patterns of surface flows. In particular, they have been able to create periodic patterns of rotating waves that can transfer local angular momentum to micro-particles leading to micro-particle trajectories in the form of spatially periodic nested orbitals.

I found the experimental (and theoretical) results very exciting and the work certainly merits for publication in NCOMMS after some mandatory revisions that are summarized below:

A: We thank the referee for his positive review and for his insights.

1) The authors state in the introductory paragraphs: “Here we show that a liquid surface metamaterial can be created by wave-controlled patterns of surface flows. The attraction of this wave approach is that it relies solely on hydrodynamic forces to guide particles”. The authors already discussed the same idea in Ref. 21 where they introduced “a new conceptual framework for understanding wave-driven flows”. The authors should clarify and emphasize the actual novelty and relevance of their results compared with their previous work.

A: By a liquid metamaterial we mean spatially periodic patterns capable of guiding matter in a way which indeed is similar to optical lattices. Even 3-4 years ago the generation of horizontal vortices on the water surface by the surface waves was not known in the fluid mechanics community. In our 2014 paper we stressed that the ability to create such vortices would allow engineering surface flows. The main difference between results of 2014 and this manuscript is in the nature and the spatial structure of those waves. In Ref 21 we used strongly nonlinear propagating waves for which a theoretical description is yet to be developed. In this paper we use orthogonal linear standing waves for which it was possible to develop a theoretical model. Standing waves form a landscape for a deterministic quasi-two-dimensional periodic flow. Crucially, this new experiment uncovers the role of the temporal phase in the generation of the vortex-like structure and provides a strong basis for an analogy with optical lattice. We now discuss these points in the Introduction and in the Discussion sections.

2) In the abovementioned reference [21] the authors used wave-controlled patterns of surface flows to develop a surface-wave analogue of an Optical Tractor Beam [It is interesting to note that while in the ArXiv version of this paper, the authors cited and explicitly acknowledged that the work was motivated by previous work on Optical Tractor Beams, in the published version in Nature Physics (Ref. 21), references to the previous work on optical forces were removed...]. In the present work, the authors discussed the flow patterns associated to superposition of orthogonal standing waves. The optical analogue of the author’s approach can be found in an early work on optical lattices by Hemmerich and Hänsch [A. Hemmerich and T.W. Hänsch, Phys. Rev. Lett. 68, 1492 (1992)] and the flow patterns of nano-particles illuminated by two crossed optical standing waves have been discussed by Saenz and coworkers [Albaladejo et al., Phys.

Rev. Lett. 102, 113602 (2009); Nano Lett. 9, 3527 (2009); Zapata et al., Phys. Rev. Lett. 103, 130601 (2009).] These references should be cited in the manuscript.

A: We appreciate this comment. We should mention that the references to optical tractor beam in our Ref.21 were removed in response to the suggestions by the referees of that paper who did not find the analogy physically justifiable. We agree that similarities with optical lattices are striking and we thank the referee for bringing to our attention these very relevant references. We have included them in the revised manuscript. We also included the comparison between wave-based liquid metamaterial and optical lattices in the Discussion section.

3) It is remarkable that, in full analogy with the Optical case, when there is a $\pi/2$ phase shift between the orthogonal standing waves, the flow patterns resemble those of a periodic vortex lattice. While the main driving forces in the optical case are proportional to the energy flow (the Poynting vector), the dynamics of the particles on the surface of the fluid, driven by the fluid flow, is more complex.

Could the authors include a brief discussion of the similarities and differences of the driving by light radiation pressure and liquid flow?

This will help the interdisciplinary readers of NCOMMS to understand better the beautiful results presented by the authors.

A: We thank the referee for this comment, which we find enlightening. The analogy between the liquid-surface metamaterials presented here and optical lattices is indeed very useful for the interdisciplinary readership. We have added the proposed references and discuss the similarities and differences between the two systems in the last part of the paper. In particular, we emphasize the difficulties in defining a wave momentum in fluid mechanics, and the special role played by the Stokes drift in that respect.

This analogy needs to be further developed. Thanks again, we believe this is a good addition to the paper.

Reviewer #2 (Remarks to the Author):

Combining experiments and theory Francois et al demonstrate and explain the orbital and drifting motions of solid particles surfing on Faraday-wave lattices.

The experiments are very elegant and yield extremely accurate results both for the interface and the (3D) particle dynamics. They rely on a host of advanced techniques mastered by the authors for the interface and flow measurements. The model is simple and quantitatively accounts for most of the experimental findings.

A: We thank the referee for his positive review and his constructive criticism.

However, I cannot recommend this manuscript for publication in the present form:

(i) The presentation of the results is often unclear, at least to nonspecialists in the field. I would strongly recommend adding to the main text some of the explanations provided in the method section. This comments especially applies to the theoretical model and to the description of the experimental setup.

A: We have restructured the paper which was originally submitted to a different journal and was restricted by a page limit. We have greatly expanded the Introduction section, and added to the main text a section on the experimental setup and the theoretical model. In particular, we have clarified the derivation of the Lagrangian drift in the text and added a new Figure 6 to illustrate the classical Stokes drift in planar progressive waves.

(ii) The main text is marred with unsubstantial statements about potential applications of this experiment. I feel that this should be rectified.

A: We have corrected that throughout the main text. The text is now focused on the physics of wave driven flows on a fluid interface, the similarities and differences with the particle motion in optical lattices, and recent ideas related to the liquid-interface metamaterials.

(iii) The following more specific points should also be addressed:

-1- Unlike what is suggested in the abstract, the dynamic control of particles by wave fields is very common. The most obvious example is provided by the optical lattices routinely used in atomic physics and colloidal science.

A: In the introduction and in the Discussion section we now acknowledge the remarkable analogy that exists between this work and studies on optical lattices. Yet, surface waves are very different from either electromagnetic or acoustic waves, and, to our best knowledge, surface waves have not been routinely used to manipulate floaters. In the revised manuscript we therefore emphasize important limitations of the analogy between surface wave phenomena

and electromagnetic waves. Such limitations are related to rather different wave dispersion relations, to the definition of a wave momentum and to the particle polarizability.

-2- A lot of emphasis is put on the manipulation of colloidal particles, yet the scale of the lattice used in this experiment is of the order of a centimeter. Positioning colloids with a millimeter precision on a square lattice with a spacing of 1 cm would be definitely easier using pipets. Moreover, capillary forces would definitely dominate the Stokes drag driving the particles. The authors should either quantitatively address these points or significantly adjust their enthusiasm about the relevance to colloid and cell manipulation.

A: In the latest version of the paper we have removed any mentioning of colloidal particles. To address the question raised by the referee on the competition between capillary forces and the hydrodynamic force induced by the Stokes drift (since our particles are fluid tracers, we presume that the referee meant Stokes drift, not Stokes drag), we now draw an analogy between the surface water wave lattice and optical lattices.

In this picture:

- The action of rotating water waves is the analogue of the force induced by a radiation pressure in electromagnetic waves.
- The wettability effects may be analogous to the dipole force in optics.

The second analogy is supported by the work of Falkovich et al.[10] where it was shown theoretically and experimentally that capillary effects in standing waves can be derived from a potential energy. In this theory, capillary forces are coupled to particle inertia and act somewhat similar to the dipole force in optical lattices: i.e. forcing them to drift to the nodes or antinodes of the standing waves according to their wettability (which plays the role of the induced dipole in optics).

In these experiments, particles are 25 microns in radius and we do not observe any accumulation at the nodes or antinodes on the timescale of our experiments. It shows that the effect of the Stokes drift is dominant.

-3- What is specific to the nonlinearity of the Faraday waves? Again, I am not an expert in interfacial waves and Lagrangian transport, reading this manuscript I could not understand what could not be achieved with standing capillary or gravity waves. Would the physics and the phenomenology be qualitatively different?

A: We think this is a misunderstanding which we tried to clarify in the revised manuscript. In this paper we do not deal with the Faraday waves. Waves in these experiments are *linear* standing waves. We have substantially rewritten the Introduction section and added a section on the experimental setup to make this clear.

Faraday waves are parametric excitations. They are unstable and easily break into ensembles of nonlinear oscillating solitons [5-7]. Transport of matter on the surface perturbed by such nonlinear waves [12-14] is indeed qualitatively different from that induced by the linear wave field described here.

Throughout the text and with the addition of a new Figure 6, we have clarified the meaning of the Lagrangian description of fluid motion as well as the Lagrangian nature of the drift observed in this experiment.

-4- The description of Eq. (2) is not correct. ξ is the particle displacement, not its instantaneous position. Galilean invariance would be broken otherwise... Eq. 2 also includes a mean flow term (which vanishes in the present case)

A: This has been corrected (see Equation (9)).

-5- Some of the panels in figures 1 and 3 are not discussed in the main text.

A: Thanks for noting that. All the panels of Figures 1 and 3 are discussed in the new subsections *Experimental Setup* and *Wave Driven Fluid Motion*.

-6- Given their importance a clear definition of the nodal points is required.

A: A definition is now given in the main text (in the section *wave driven fluid motion*); nodal points are points of zero surface displacement where the local amplitude of the standing wave field is zero at every instant in time.

-7- When does particle inertia becomes relevant (particle size and oscillation frequency)?

A: This is an interesting and difficult question that we are currently exploring experimentally. The difficulty comes from the interplay between capillary forces and particle inertia (size, density) when a particle is at an interface (see Falkovich et al. [10]), an effect that we have already mentioned in our answer to question 2.

In this work we use small (radius=25 microns) imaging particles with a density close to that of water (particle density =1.03 g.cc⁻¹). The good match with the theoretical model confirms they are true fluid tracers and they are not affected by inertial effects.

-8- The theoretical explanation for the rotation drift term should be explain in more details. This is one of the main result of the experiment, as far as I understand it, Eq. 12 seems to be established on a sole phenomenological basis. Is it correct?

A: Yes, the drift U^R is established on a phenomenological basis. The Stokes drift is a Lagrangian effect computed for linear waves (corresponding theoretically to infinitely small amplitude waves $KH \ll 1$). The drift U^R is introduced to take into account a higher order effect of the finite steepness of the waves in our experiments.

-9- I found the discussion of the "collective" properties rather confusing. What is collective here? As far as I understand this last section, only the flow field is characterized, not the particle positions. The particles are used as passive tracers, and do not interact. I do appreciate the characterization of the nonlinear response of the flow, but do not see any sign of a collective phenomenon.

A: The use of the term "collective properties" is indeed confusing. It has been removed.

-10- The authors might want to refer to this recent paper;
Fluids by design using chaotic surface waves to create a metafluid that is Newtonian, thermal, and entirely tunable, by Welch et al, P.N.A.S 2016.

A: Thanks, we now cite this reference in the Introduction and in the Discussion.

REVIEWERS' COMMENTS:

Reviewer #1 (Remarks to the Author):

The authors have considered all the points raised in my previous report providing a satisfactory answer. From my point of view, they have also answered most of the questions of the other reviewer. The revised manuscript has been improved significantly and I recommend its publication in NCOMMs.

Reviewer #2 (Remarks to the Author):

The authors have answered all the questions raised buy the referees and have clarified the main text. The discussion of the analogies and differences with optical lattices suggested by the other referee is a nice addition to the manuscript.

I recommend the publication of the article in Nature Communications as is.